# Prevalence and Risk Factors Associated with Mental Health in Adolescents from Northern Chile in the Context of the COVID-19 Pandemic

**DOI:** 10.3390/jcm12010269

**Published:** 2022-12-29

**Authors:** Rodrigo Moya-Vergara, Diego Portilla-Saavedra, Katherin Castillo-Morales, Ricardo Espinoza-Tapia, Sandra Sandoval Pastén

**Affiliations:** 1Escuela de Psicología, Facultad de Humanidades, Universidad Católica del Norte, Antofagasta 1240000, Chile; 2Escuela de Psicología, Facultad de Ciencias Sociales, Universidad Santo Tomás, Antofagasta 1240000, Chile

**Keywords:** anxiety, Chile, COVID-19, depression, social phobia

## Abstract

The COVID-19 pandemic has affected the world population; however, there is limited knowledge about its impact on adolescents. The aim of this study was to estimate the prevalence and risk factors associated with mental health in the context of the COVID-19 pandemic in young people in northern Chile. The sample consisted of 1315 young people between the ages of 14 and 18. Univariate analysis and multiple logistic regression were performed to identify the risk factors associated to the considered symptomatology. Depressive symptomatology was reported at 36.3%, anxiety at 6%, and social phobia at 27.8%. Adolescent females reported a higher probability of presenting depressive (OR, 3.409), anxious (OR, 3.949), and social phobia (OR, 3.027) symptomatology. Young women between the ages of 17 and 18 were more likely to present anxiety (OR, 2.172); moreover, being a migrant was related to lower odds of presenting anxiety (OR, 0.259) and depression (OR, 0.662). Having a family member diagnosed with COVID-19 was associated with depressive symptomatology (OR, 1.369). The COVID-19 pandemic can have serious repercussions on the mental health of adolescents. It is important to generate interventions to meet the needs of youth during the post-confinement period.

## 1. Introduction

The COVID-19 pandemic caused the vast majority of the world’s governments to implement measures in order to contain the spread of the virus [1]. This was done to protect the health of their citizens, both the adult and youth populations [2,3,4,5], the latter being those who presented a lower percentage of diagnosed cases [6,7,8]. Some of the measures implemented were the mandatory use of masks in public spaces, physical distancing, closing of educational establishments and borders, and partial or total quarantines in homes, which were often prolonged over time [9,10,11].

In this sense, it has been pointed out that sudden and drastic interruptions in the daily life of young people during the pandemic may have caused adverse effects in their mental health [12,13,14,15]. This situation is troublesome, considering that young people in general present greater mental health problems [16].

Presently, evidence of the medium- and long-term psychological impact of the pandemic is still being generated. Studies have focused more on the impact of COVID-19 on the mental health of the adult population [17] compared to the youth population [18], where there is little evidence [19]; specifically, this evidence is scarce in the adolescent population.

Some studies have identified mental health problems in young people [20,21,22], highlighting an increase in depressive and anxious symptomatology [18,21,23,24], generalized anxiety, social anxiety [12], sleep disorders [25], and post-traumatic stress symptoms [20].

Furthermore, Ipsos [26] in Chile conducted a survey on mental health during the pandemic. The results suggest that 56% of the Chilean population think that mental health has worsened as a result of the health emergency. In addition, a study conducted by Urzúa et al. [27] points out that extreme fear and uncertainty were frequent responses in the Chilean population before the pandemic. People would also present symptomatology of anguish, insomnia, anger, social isolation, and/or specific symptomatology of post-traumatic stress disorder, depression, and anxiety. In addition, in a survey carried out by the National Institute of Youth [28] it was found that 54.4% of adolescents between the ages of 15 and 19 have felt very or quite stressed since the beginning of the pandemic, a concerning situation, considering that according to UNICEF [24], Chile has decreased the number of public mental health care facilities for children and adolescents since 2019.

### Mental Health Risk Factors

The psychological effects of COVID-19 can increase when young people pose different risk factors. Some of these have been associated with sociodemographic variables such as gender [29], age [30], migration [31], or socioeconomic levels [32]. In addition to this, Retamal [33] states that the context of the COVID-19 pandemic compromises individuality and collectivism in young people. This could imply that experiencing some social contact during COVID-19, such as being diagnosed, or a significant person having some issues related to the illness, in some way influenced the mental health of a person during the pandemic. However, this is not entirely clear. For example, a study found that social contact during COVID-19 was not associated with the mental health of people, as there were other variables of greater interest that were not considered in this research [34].

Most of the research carried out on the impact of COVID-19 on the mental health and psychological well-being of young people has been conducted in high-income countries [35], and few studies have been done in developing countries such as Chile. There is a theoretical vacuum in the research with regards to the prevalence and risk factors of mental health in young people during the pandemic and specifically in the adolescent population. This situation, as a by-product of the measures taken to contain the contagion, is worrisome, as the Committee on the Rights of the Child [36] has expressed its concern that this could have harmful effects on those vulnerable at the physical, emotional, and psychological level. Thus, Orgiles et al. [37] state that approximately 860 million children and young people in the world could have been affected by the measures implemented by the quarantine.

The present research aims to estimate the prevalence and risk factors associated with mental health in young people in northern Chile during the COVID-19 pandemic. The importance of this study is that Chile has a high prevalence of mental health issues in its population. However, services that address these problems are scarce [38], especially for adolescents, where Chile has decreased mental health benefits to girls, boys, and young people [24]. Thus, conducting studies on the prevalence and risk factors of the effects of the pandemic on the mental health of young people raises awareness and guides public policies. Consequently, it is important to invest in and promote mental health programs that favor the reduction of the symptoms of this population and thus avoid its chronicity over time.

## 2. Materials and Methods

### 2.1. Participants

This study was non-experimental and cross-sectional. Through non-probabilistic sampling, a questionnaire was applied in-person and online. The sampling technique was by convenience, in order to access as many participants as possible, according to the inclusion criteria.

The final sample of the study consists of 1315 young people between the ages of 14 and 18, from different communities in northern Chile; the average age is 16 years (SD = 1.26). The inclusion criteria were to reside in any of the communes of the Antofagasta region in Chile, to be between 14 and 18 years of age, and to not have any cognitive disability to be able to adequately answer the consent form and the respective survey. There was no specific hypothesis about this particular geographic area; it was only necessary to characterize the phenomenon in the area, i.e., this inclusion criterion was considered because it represents the range of action where the sponsoring institution, the Observatory of Children and Youth of the Catholic University of the North, operates. The exclusion criteria were a cognitive or volitional disability that prevented potential participants from giving their consent effectively. This criterion was considered, taking into account the educational establishments that were contacted in Chile, where young people who do not have severe cognitive disabilities attend.

### 2.2. Procedure

In order to obtain locations that complied with the required age range, that is, adolescents between the ages of 14 and 18, community authorities and directors of educational establishments in the region of Antofagasta, Chile, were contacted. The participants signed an informed consent form. The application of the instruments was carried out between the months of September and December 2021. In all cases in which the educational establishments gave their approval to participate, the survey was distributed uniformly, giving priority to the school levels in which the adolescent population in the required age range could be studying. The participating adolescents accessed the questionnaire at their schools or through a link to the questionnaire available online. In the case of a face-to-face application, the instruments were applied to the young people with a trained interviewer. The estimated application time was between 40 and 60 min. In total, 1550 instruments were applied in the 9 communities and in different educational establishments in the region. Of these, 1315 were considered for the final analyses. Some questionnaires were discarded as they were incomplete, or despite being answered, the consent section expressly stated the refusal to participate. The instruments used had as their main requirement that the participant should be able to refer to the last month/week in temporal terms. This was in order to avoid recall bias. For example, some of the sections of the survey had as a prompt, “Please indicate how often the following things happen to you, as a result of the pandemic period”, or even in the last week, “Using the scale below, indicate the alternative that best describes how often you have felt that way during the last week”.

Finally, given that the questionnaires asked about aspects related to mental health symptoms during the pandemic, a protocol for psychological support and accompaniment was developed for these situations. This protocol included the analysis of the most relevant mental health scores found in the questionnaire. When the score was high in some of the mental health symptomatology, contact was established with the adult responsible for the participant to offer information about the psychosocial care center of the Universidad Católica del Norte. This was done so that he/she could receive psychological care or support if necessary. 

In addition, this research complied with the principles of the Declaration of Helsinki. The present investigation has an approval letter from the ethics committee of the Universidad Alberto Hurtado, Chile. This was approved on 2 March 2021. This committee does not work with a numerical code of approval. However, both the letter of approval and the respective informed consents used were signed and stamped with the different seals of the respective committee.

### 2.3. Instruments

The Sociodemographic Information Questionnaire was used to collect information regarding gender, age, community, type of educational establishment, educational level, and migration circumstance. Additionally, inquiries were made about issues linked to COVID-19 (e.g., family members who have died due to COVID-19, knowledge of a significant person dying).

The State–Trait Anxiety Inventory (STAI) was used to measure two dimensions of anxiety: state and trait [39]. For this research, a version based on baseline data from the Chilean population was applied [40,41]. Accordingly, 20 anxiety items were used as state dimension. Participants reported at what level each statement reflects how they felt on the day the questionnaire was administered. Participants responded to the different items by indicating 0 (not at all), 1 (somewhat), 2 (quite a lot), and 3 (very much). Some examples of items are: I feel nervous; I feel very oppressed; I am worried about possible future misfortunes. The state dimension of anxiety version has exhibited adequate psychometric properties in several studies with the Chilean population [42,43]. This study reported an α = 0.92, indicating high reliability.

The Center for Epidemiological Studies Depression Scale (CES-D) instrument was used, consisting of 20 items, which correspond to usual and representative symptoms of depressive disorder [44]. Four items (4, 8, 12, and 16) needed to be inverted, as they are presented in a positive way. The prompt asked participants to indicate the frequency with which the different symptoms were experienced; responses ranged from 0 (1 day or less), 1 (one time or few times), 2 (occasionally or several times) and 3 (most of the time). Some examples of items are: I thought my life had been a failure; I felt that I could not stop from being sad, even with with the help of my family and friends; I felt little desire to eat; I had a bad appetite. This instrument has exhibited adequate psychometric properties in national adaptation and validation studies [45,46]. For the present research, the instrument reported an α = 0.90, indicating high reliability.

The Social Avoidance and Discomfort Scale is composed of 28 items, of which both the subjective discomfort produced by situations of social interactions and the avoidance or desire to avoid them are evaluated [47,48]. The prompt suggests answers that include two options, true (1) and false (0). Some examples of items are: I try to avoid situations where I have to be very sociable; I usually get nervous about being around people, unless I know them; I often want to run away from people. This instrument has exhibited adequate psychometric properties in review studies, being used as an instrument to measure symptoms of social anxiety [49,50]. In addition, this questionnaire has been used in a study with a Chilean population, showing adequate psychometric properties [51]. For this research, the instrument reported an α = 0.90, which indicates high reliability.

### 2.4. Statistical Analysis

Initially, a univariate analysis was performed. That is, percentages were estimated for the characterization of the sample, the variables related to the COVID-19 pandemic, and the levels of symptomatology present in the participants. This was done by excluding missing values in each of the items of interest. 

In the second stage of analysis, and with the purpose of conducting a multivariate analysis, two levels were established for each symptomatology, low and high, based on the cut-off scores provided in the literature for social phobia, anxiety, and depression [45,49,52], when these are in clinically significant ranges. For anxiety symptomatology, the cut-off score was 40 points; for depression, the cut-off score was 24 points; and for social phobia, it was 17 points. Subsequently, a Chi-square analysis was performed to estimate the association between the sociodemographic and COVID-19-related categorical variables and the different levels of symptomatology previously described. 

The identification of these associations made it possible to decide to apply a multivariate logistic regression model in order to explore the factors potentially associated with symptoms of depression, anxiety, and social phobia. Logistic regression was used, since this type of analysis predicts the outcome of a categorical variable (symptom level) based on the independent or predictor variables [53]. In this study, the independent or predictor variables used were gender, age, migrant status, geographic area of residence, type of educational establishment, and the four questions regarding COVID-19 (i.e., (i) Have you been diagnosed with COVID-19? (ii) Have any members of your family, with whom you live, been diagnosed with COVID-19? (iii) Have any members of your family died from COVID-19? (iv) Any significant people who have passed away from COVID-19?). Three logistic regression models were carried out for the three types of symptomatology (anxious, depressive, and social phobia). 

In the case of gender, this variable was dichotomized into male and female, due to the small number of participants who identified themselves as non-binary. Thus, the odds ratios (OR) and the 95% confidence intervals are presented. The level of significance was set at *p* < 0.05. All statistical analyses were performed with the statistical software SPSS version 25.

## 3. Results

Table 1 shows the frequency distribution of the sample in relation to different sociodemographic characteristics.

As shown in Table 1, 49.8% of the participants identified themselves as female, 48.4% as male, and 1.8% as non-binary. In the types of educational establishments, 53.7% declared attending public schools, 22.1% subsidized, 15.3% private, and 2.2% universities or institutes of higher education. Regarding the educational level, 31.5% said they were in eighth grade (primary), 25.5% in first year of secondary, 20.5% in second year of secondary, 16.4% in third year of secondary, and 4.3% in fourth year of secondary. On the other hand, only 2.1% said they were studying the first level of tertiary education. With regard to nationality, 86.7% participants identified themselves as Chileans. The nationality with the second-greatest number of participants is Bolivian with 6%, followed by Colombian with 3.6%.

### 3.1. Demographic Characteristics Related to the Pandemic

As shown in Table 2, 10.4% of the participants were diagnosed with COVID-19, although the majority did not have this diagnosis (89.6%). For their part, 29.1% of participants reported that a family member had been diagnosed, and 8.8% that a relative had died from COVID-19. Additionally, 21.5% of participants indicated the loss of significant people as a result of the pandemic.

Table 3 shows the percentages of symptoms reported by the participants. Regarding anxiety, 6% of the participants reported a high symptomatology; for depression, more than a third of the sample (36.3%) indicated this symptomatology in a high range. In the case of symptoms of social phobia, 27.8% reported this condition at a high level.

It was decided to perform a complementary Chi-square analysis to measure the level of association of the qualitative variables previously mentioned. Table 4 shows sociodemographic variables associated with depressive symptomatology in the participants. In this regard, gender was found to be significantly associated with depressive symptomatology (low, medium-high) [χ^2^ = 97.46, *p* < 0.05]. In the case of migrant status, a significant association with depressive symptomatology was also found [χ^2^ = 4.15, *p* < 0.05]. Finally, the type of educational establishment also had a significant association with depressive symptomatology [χ^2^ = 8.48 *, *p* < 0.05]. Regarding the specific variables related to COVID-19, significant associations were found with depressive symptomatology and between having a diagnosed family member [χ^2^ = 11.87, *p* < 0.05], a deceased family member [χ^2^ = 4.94, *p* < 0.05], and a deceased important person [χ^2^ = 5.92, *p* < 0.05], all with a cause associated with COVID-19.

Regarding the association between the previously described variables and anxiety symptomatology, significant associations were found between the sociodemographic variables and anxiety symptomatology, that is, gender [χ^2^ = 25.53, *p* < 0.05], age [χ^2^ = 11.54, *p* < 0.05], migrant status [χ^2^ = 8.10, *p* < 0.05], and type of educational establishment [χ^2^ = 8.01 *, *p* < 0.05]. Regarding variables directly related to COVID-19, no significant associations were found (*p* > 0.05). Finally, in reference to social phobia symptomatology, significant associations were found with gender [χ^2^ = 69.11 *, *p* < 0.05] and with the variable referring to whether any family member had been diagnosed with COVID-19 [χ^2^ = 5.74 *, *p* < 0.05]. The rest of the sociodemographic variables in relation to COVID-19 did not show statistically significant associations.

### 3.2. Factors Associated with Symptoms of Depression, Anxiety, and Social Phobia

The results of the sociodemographic analysis and pandemic-related variables are presented in Table 5. The Hosmer and Lemeshow test was performed for the three logistic regression analyses for the symptomatology of depression (χ^2^ = 8.515, *p* = 0.38), anxiety (χ^2^ = 6.069, *p* = 0.63), and social phobia (χ^2^ = 2.805, *p* = 0.94). In all three models, an adequate fit is visualized.

The multivariate analysis revealed that females showed a significantly increased depressive symptomatology (OR, 3.409; 95% CI, 2.61–4.44), increased anxiety (OR, 3.949; 95% CI, 2.15–7.24), and increased social phobia (OR, 3.027; 95% CI, 2.26–4.04). That is, females were more likely than males to report symptoms in increased ranges during the pandemic. On the other hand, in terms of age, participants between the ages of 17 and 18 reported a greater probability of suffering from increased anxiety than participants between the ages of 14 and 16 (OR, 2.172; 95% CI, 1.29–3.65). Additionally, associations were identified between migrants and symptoms of depression and anxiety. In this sense, young migrants were less likely to present symptoms of increased depression (OR, 0.662; 95% CI, 0.45–0.95) and increased anxiety (OR, 0.259; 95% CI, 0.78–0.85). Regarding specific variables, the only significant one in the model was the effect of recognizing a relative who was diagnosed with COVID-19, showing an increase in depressive symptomatology (OR, 1.369; 95% CI, 1.00–1.86). The remaining variables were not statistically significant. Finally, the model for depressive symptoms explained 13% of the variance, for anxiety 12%, and for symptoms of social phobia 9%.

## 4. Discussion

The objective of this study was to estimate the prevalence and risk factors associated with mental health in the context of the COVID-19 pandemic in adolescents in northern Chile. The results show that 10.4% of the participants had been diagnosed with COVID-19 at the time of the study. This finding is directly related to other international and national studies that show the low prevalence of COVID-19 in this population [6,7,8]. However, it is necessary to highlight that this study considered a sample for convenience and is not probabilistic; thus, our findings are not representative, but only demarcate a trend.

On the other hand, 29.1% of the participants reported that a family member had been diagnosed with CO-VID-19 prior to the study, while 8.8% indicated that a family member had died from this disease. This is related to studies conducted in Spain, where it was found that 13% stated that a family member had been diagnosed, and 8.2% stated that a family member or a significant person had died from the virus [34]. This was relevant to the present study, since during disease outbreaks, members of the family system may experience anxiety and fear about the possibility of becoming infected [54]. In this sense, unpredictability due to the spread of COVID-19 could have contributed to the uncertainty in many family systems and to presenting depressive and/or anxious symptoms, due to the fear of catching the disease or consequences that it could generate in infected relatives [55].

Regarding the mental health indicators addressed in this study, the findings are consistent with other research that reports the increase in depressive and anxious symptoms in young people during the pandemic [13,18,20,23,25,27]. Regarding the adolescent population, studies that have been proposed to analyze the symptomatology presented in this age group have generally been conducted in developed countries. For example, studies in China [56,57,58], in the United Kingdom [59], and USA [60] propose the existing relationship between the pandemic context and anxious symptomatology. A similar scenario arises when analyzing depressive symptomatology in adolescents during the pandemic. Some studies reveal the association of this type of symptomatology with variables specific to the pandemic context, although these studies had the same characteristics as those previously mentioned, i.e., they were generally conducted in more developed countries [57,61,62]. In this sense, our results contribute to the empirical gap in the analysis of adolescent symptomatology during the pandemic in a developing country such as Chile. In fact, there is practically no research on the subject in countries with these characteristics [13]. In this field, we found that 6% of adolescents reported anxious symptomatology and 36.3% depressive symptomatology, in both cases in high ranges. It should be noted that a tentative explanation for this situation may be related to the restrictive measures implemented by each government to reduce the contagion in its population. Thus, for young people experiencing these restrictions, this could be complex since they depend, to a great extent, on connections and social interactions with their friends, who constitute and are configured as primary emotional support [18]. This could have affected their mental health, including having no direct social contact during COVID-19 [34].

In relation to the social phobia addressed in this research, 27.8% of adolescents reported this symptomatology at a high level. It is important to highlight that it is a variable that has been scarcely studied. The pandemic itself has implied abrupt transformations of the social schemes and routines to which people were accustomed [18]. That is, even dispensing with a direct link with the COVID-19 disease, its implications in the social and interpersonal scenario are unavoidable [63]; it appears that the symptomatology of social phobia is transversal in the adolescent population, given the high percentage reported.

Another issue of relevance to the present investigation was the associated risk factors. In this sense, we found significant associations between gender, migrant status, type of educational establishment, and COVID-19 diagnosis of a family member, and depressive symptomatology. In the case of anxious symptomatology, we found associations of gender, age, type of settlement, and migration status. However, no associations were found with this symptomatology and variables related to COVID-19. With respect to social phobia, only gender and the diagnosis of a family member with the disease were significant. This allows us to reflect on the fact that there are generally sociodemographic factors that were mostly related to symptomatology during the pandemic. Moreover, only some situations of social contact during COVID-19 seemed to be related to the symptomatology analyzed.

Taking these relationships into account, we decided to perform a multivariate binary logistic regression, which allowed us to analyze these associations in greater depth. For example, our results suggest that adolescent females would be more likely to present high levels of depressive, anxious, and social phobia symptomatology, being consistent with other studies available in the literature [23,29,30,64]. This finding can be explained from the process of gender socialization and the sociocultural construction of expectations, rules, and duties linked to the female role, which, in the context of the pandemic, could have worsened within the family, resulting in greater responsibilities and/or increased domestic work [31]. This would be added to the duties of the age range, such as school or virtual classes taught during the pandemic.

Another risk factor of interest was the age range of the adolescents. In particular, in the present study, adolescents who were in the age range between 17 and 18 were more likely to have high anxiety symptomatology, unlike adolescents who were in the age range of between 14 and 16 years old. An explanation for this result may be the product of personal and/or social pressure regarding the new educational and/or work challenges that young people are soon to face after finishing secondary education, the foregoing considering the loss and deterioration of educational learning during 2020 [65]. This could mean that young people in this situation perceive themselves as less self-sufficient and more insecure in the face of these new challenges. This is consistent with the literature, which has related being a student with a higher psychological impact and with anxious symptomatology during the pandemic [13,30].

Immigration status was another risk factor considered. The literature indicates that being a migrant could generate greater affectation during the pandemic; this is due to the fact that the migrant population has higher levels of poverty, overcrowding, precarious housing conditions, labor informality, and job losses [31]. However, contrary to what was previously stated, the present study shows that young migrants had a lower probability of presenting symptoms in high ranges compared to Chileans. In this sense, it should be noted that the "migrant" factor was made up of young children of immigrant parents who were already established, children who were settled and enrolled in school in the country, by which they could feel greater stability in facing COVID-19 [66], understanding this as feeling capable of coping with difficulties at an individual, family, and community level [67]. This perception of stability could be associated with presenting fewer anxious and/or depressive symptoms during the health crisis [66]. Regarding the Chilean population in general, this finding ratifies the results of previous research, with respect to Chile being a country with a high incidence of mental illnesses and poor accessibility to mental health services [38].

On the other hand, and as mentioned regarding the interest in analyzing social contacts with COVID-19, it was observed that living with a relative who had been diagnosed with the disease was associated with high depressive symptomatology in young people. This result is contradictory to a previous study, which suggests that contagion from relatives or close people would not be related to the psychological impact or mental health problems in adults [34]. A plausible explanation for these differences may be the product of age, since the present study was made up of young people between the ages of 14 and 18, while other research was conducted with people over 18 years of age. Generally, the adolescent population maintains greater susceptibility to different vital changes, a product of the transformative implications at the psychological, emotional, and behavioral levels at this stage of development, in addition to the relevance of maintaining family and interpersonal relationships in a satisfactory range [68].

Finally, both the socioeconomic level and the other variables of interest related to COVID-19 were not significant, which opposes the research that has proposed a more direct relationship between these variables. For example, consider the study by Shi et al. [69], who found that being diagnosed with COVID-19 was related to higher symptoms of depression and anxiety in the population. However, these differences can also be explained because the aforementioned study was conducted in the country where the pandemic originated, that is, in China in the year 2020, unlike the present study, which was carried out during the second half of the year 2021. For its part, it should be noted that the Chilean Ministry of Health, in order to deal with the pandemic, assumed a risk communication policy in order to promote self-care and awareness measures in general [70]. This could have implied a greater knowledge of the causes and implications of the disease among the Chilean population, which could have translated into better strategies to deal with both own’s one diagnosis and that of a member of the close circle.

### 4.1. Practical Implications

The results of this study may have important practical implications for the creation of intervention programs for the youth population in Chile who have been exposed to the harmful effects of the pandemic. In addition, the results can be transferred in a formative way to professionals who work directly with young people in educational establishments or psychosocial programs. Although the analysis of the figures in this study can help the authorities become aware of the seriousness of the problem, it is also important to provide families and young people with positive coping strategies for the effects of the pandemic, specifically in these post-confinement times. Regarding the results on risk factors, these can be useful for authorities, for example, organizations such as the Ministry of Health, the Ministry of Education, and the Ministry of Social Development of Chile to implement promotion and prevention plans and programs. Resources could be optimized by carrying out focused prevention with the vulnerable groups identified in this research.

### 4.2. Limitations and Future Research Directions

This research is not without limitations. By having a non-probabilistic sample and for convenience, the data presented here are not representative. Furthermore, whether the person had mental health symptoms prior to the pandemic was not quantified, which is assumed to be a limitation of the research, typical of cross-sectional designs, and this could also have altered our findings. However, they reflect a trend and a first approach to the mental health situation of young people in northern Chile. For its part, the investigation was carried out during the second half of 2021, during a time of greater management by the authorities of infections and the repercussions of the pandemic, circumstances that could have affected the results presented.

Future research on the matter could delve into specific mental health issues such as social phobia, which has been scarcely addressed in the literature, in addition to expanding the age range to the population of boys and girls, which have been relegated to the background in investigations of these characteristics. It would be convenient to conduct research in other geographical latitudes in Chile, or indeed, to carry out longitudinal research that allows the mental health of young people to be measured at different stages during the post-confinement period.

## 5. Conclusions

The prevalence of symptoms of depression, anxiety, and social phobia in young people from northern Chile was high during the COVID-19 pandemic. Our study found a higher percentage of depressive symptoms and social phobia. Being female related to all three symptoms, being between the ages of 17 and 18 with anxiety, and being Chilean with anxiety and depression. In the case of the variables related to COVID-19, only having a relative diagnosed with the disease was associated with high depressive symptomatology. These results suggest that the COVID-19 pandemic may have serious repercussions on mental health. Future studies are required to explore the association between other variables of the pandemic, such as changes in social activities, family and school routines of young people, and their implications for mental health. Finally, specific interventions based on the knowledge generated from these studies are urgently needed to meet the needs of young people during the post-confinement period.

## Figures and Tables

**Table 1 jcm-12-00269-t001:** Sociodemographic characteristics of the participants.

		N	%
Gender	Female	654	49.8%
	Male	636	48.4%
	Non-binary	24	1.8%
Age	14 years	295	22.4%
	15 years	322	24.5%
	16 years	313	23.8%
	17 years	279	21.2%
	18 years	106	8.1%
Communities	Antofagasta	662	50.3%
	Calama	259	19.9%
	Taltal	172	13.1%
	María Elena	99	7.5%
	Tocopilla	89	6.8%
	Mejillones	13	0.9%
	Sierra Gorda	13	0.9%
	Ollagüe	8	0.6%
Type of Educational Establishments	Municipal	706	53.7%
	Subsidized	291	22.1%
	Private	201	15.3%
	Information unknown	88	6.7%
	University/Institute	29	2.2%
Educational Level	8th Primary	57	31.5%
	I Secondary	414	25.2%
	II Secondary	269	20.5%
	III Secondary	331	16.4%
	IV Secondary	216	4.3%
	1st Higher Ed.	28	2.1%
Nationality	Chilean	1140	86.7%
	Bolivian	80	6%
	Colombian	47	3.6%
	Peruvian	20	1.5%
	Venezuelan	13	1%
	Other	10	0.8%
	Argentinean	4	0.3%
	Haitian	1	0.1%

Note: The difference in sample sizes in each item are due to the omission of responses by the participants.

**Table 2 jcm-12-00269-t002:** Descriptive statistics and pandemic-related information for the total sample.

Factors	Participants (%)
Have you been diagnosed with COVID-19?	
Yes	(10.4)
No	(89.6)
Have any members of your family, with whom you live, been diagnosed with COVID-19?	
Yes	(29.1)
No	(70.9)
Have any members of your family died from COVID-19?	
Yes	(8.8)
No	(91.2)
Do you know any other significant people who have passed away?	
Yes	(21.5)
No	(78.5)

**Table 3 jcm-12-00269-t003:** Descriptive statistics and information related to the pandemic for the total sample.

Symptomatology	Participants (%)
Anxiety	
High	6
Low	94
Depression	
High	36.3
Low	63.7
Social Phobia	
High	27.8
Low	72.2

**Table 4 jcm-12-00269-t004:** Association of sociodemographic and COVID-19-related variables with symptomatology levels.

	Depression			Anxiety			Social Phobia		
	Participants (%)			Participants (%)			Participants (%)		
Variables	Low-Normal	Moderate-High	χ^2^	Low	High	χ^2^	Low	High	χ^2^
Gender									
Male	78	22	97.46 *	97.6	2.4	25.53 *	83.6	16.4	69.11 *
Female	51	49		90.9	9.1		62.4	37.6	
Age									
14–16	64.4	35.6	0.60	95.4	4.6	11.54 *	72.6	27.4	0.28
17–18	62.1	37.9		90.5	9.5		71.2	28.8	
Migrant									
Yes	69.6	30.4	4.15 *	97.9	2.1	8.10 *	73.5	26.5	0.21
No	62.4	37.6		93	7		71.9	28.1	
Place of residence									
Urban	63.4	36.6	0.77	93.7	6.3	1.47	72.4	27.6	0.22
Rural	67.6	32.4		96.5	3.5		70.3	38.7	
Type of Educational Establishment									
Public	66.5	33.5	8.48 *	95.4	4.6	8.01 *	73.1	26.9	4.26
Subsidized	56.7	43.3		90.8	9.2		69.5	30.5	
Private	65.8	34.2		92.4	7.6		78.1	21.9	
Have you been diagnosed with COVID-19?									
Yes	58.6	41.4	1.65	90.9	9.1	2.39	73.5	26.5	0.12
No	64.3	35.7		94.3	5.7		72	28	
Have any members of your family, with whom you live, been diagnosed with COVID-19?									
Yes	56.5	43.5	11.87 *	92.5	7.5	1.18	67.5	32.5	5.74 *
No	66.8	33.2		94.5	5.5		74.2	25.8	
Have any members of your family died from COVID-19?									
Yes	54.1	45.9	4.94 *	95.5	4.5	0.54	72.2	27.8	0.00
No	64.7	35.3		93.8	6.2		72.1	27.9	
Do you know any other significant people who have passed away?									
Yes	65.4	34.6	5.92 *	94.2	5.8	.04	74	26	0.58
No	57.4	42.6		93.9	6.1		71.6	28.4	

Note. χ^2^ = Chi square, * *p* < 0.05.

**Table 5 jcm-12-00269-t005:** Multivariate logistic regression analysis of risk factors associated with symptoms of depression, anxiety, and social phobia.

	Depression		Anxiety		Social Phobia	
Variables	B	OR (95% CI)/R^2^ = 13%	B	OR (95% CI) R^2^ = 12%	B	OR (95% CI) R^2^ = 9%
Gender (female)	1.22	3.409 [2.61, 4.44] ***	1.37	3.949 [2.15, 7.24] ***	1.10	3.027 [2.26, 4.04] ***
(cat. ref: male)						
Age (17–18)	0.07	1.078 [0.80, 1.44]	0.77	2.172 [1.29, 3.65] **	0.02	1.020 [0.74, 1.39]
(cat. ref: 14–16)						
Migrant	−0.41	0.662 [0.45, 0.95] *	−1.35	0.259 [0.07, 0.85] *	−0.09	0.907 [0.61, 1.33]
(cat. ref: No migrant)						
Place of residence (rural)	−0.16	0.848 [0.52, 1.38]	−0.16	0.852 [0.28, 2.56]	0.06	1.066 [0.64, 1.75]
(cat. ref: Urban)						
Type of Educational Establishmen (cat. ref: public)						
Subsidized	0.27	1.318 [0.95, 1.81]	0.57	1.771 [0.98, 3.19]	0.02	1.025 [0.72, 1.44]
Private	0.07	1.077 [0.73, 1.57]	0.63	1.884 [0.91, 3.86]	−0.25	0.779 [0.51, 1.18]
Have you been diagnosed with COVID-19?	−0.13	0.876 [0.55, 1.38]	−0.24	0.785 [0.31, 1.95]	−0.36	0.693 [0.42, 1.14]
(cat. ref: No)						
Have any members of your family, with whom you live, been diagnosed with COVID-19?	0.31	1.369 [1.00, 1.86] *	0.21	1.244 [0.69, 2.23]	0.29	1.348 [0.97, 1.87]
(cat. ref: No)						
Have any members of your family died from COVID-19?	0.33	1.392 [0.87, 2.22]	0.01	1.017 [0.37, 2.75]	0.02	1.027 [0.61, 1.72]
(cat. ref: No)						
Any significant people who have passed away from COVID-19?	0.32	1.37 [0.99, 1.90]	−0.12	0.887 [0.45, 1.72]	−0.09	0.911 [0.64, 1.29]
(cat. ref: No)						

Note: R^2^ = R square of Nagelkerke; *** *p* < 0.001; ** *p* < 0.01; * *p* < 0.05. The formula used was: $P\left(Y\right)=1/1+e^{-\left(b_{0}+b_{1}X_{1}+b_{2}X_{2}+\cdots +b_{n}X_{n}\right)}$.

## Data Availability

The data are not publicly available due to restrictions imposed by the scientific ethics committee referred to in the “Materials and Methods” section.

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
