# Peer review of "Prevalence and Risk Factors Associated with Mental Health in Adolescents from Northern Chile in the Context of the COVID-19 Pandemic"

_jcm, 2022, doi:10.3390/jcm12010269_

Round 1
Reviewer 1 Report
The topic of the current manuscript is essential to be addressed as it has a significant implication for the health of the adolescent community.
Overall, the authors addressed the main research aims with clarity.
However, there are some comments that need to be addressed:
- The sampling technique used needs further clarification and explanation, how those participants were approached?
-I suggest moving the subheading related to the procedure after the participants as it describes the study population.
-What were the inclusion and exclusion criteria? what about those who were already diagnosed with any mental health disorders or receiving any medications or have chronic illnesses, were they excluded? as all these might confound the observed results.
- there are some results presented in the methods section, suggest moving them into the results section. Move the average age of the participants and Table 1 into the results.
- what were the variables used for adjustment in multivariate analysis?
-in Table 4, what do you mean by R2 , add it to the footnote of the table.
Author Response
Dear reviewer:
Along with greetings, hoping you are doing very well. First of all, we would like to thank you for the review of the manuscript submitted. In consideration of this, we reviewed your comments and made modifications to the manuscript. The comments and associated modifications are presented in the paper. In the manuscript these changes are highlighted by Word's "change control". Thank you once again for your feedback and time.
Sincerely
Dr. Rodrigo Moya-Vergara

Reviewer 2 Report
This paper exams the prevalence and risk factors associated with mental health during Covid-19 pandemic in Northern Chile through a survey in educational establishments. The paper is interesting and well organized. Here are my comments.
Because the design and distribution of a survey will affect the results, here is the regression results primarily. Although the authors mentioned several instruments, the research questions should be given at least as a format of sample. Also, details about how to distribute the survey should be illustrated, for example, did this survey cover all young population from 14-18 years old in all schools in the northern area, or just evenly distributed the survey?
Another issue is from the Table 2, the percentages of young people who are diagnosed with Covid-19 or their families who are diagnosed or dead from Covid-19 are very low, how did the authors justify this as the mental health issues may not be associated with Covid but just because of some other reasons?
Line 99 has been illustrated in line 92, no need to repeat.
Author Response

(The authors gave the same response as above.)

Reviewer 3 Report
This study collected 1550 participants in adolescents from north Chile at later 2021 to study the mental health prevalence and risk factors. This is an important study to help with the interventions for the youth during post-confinement.
I have several comments.
1 .For the multiple logistic regression model, the author didn't provided p-value for each variable. Would you please add this to the analysis? And may be the coefficient also.
2.It would be good if the author provides formula for the multiple logistic regression analysis.
3.Could you expand how you define symptom levels based on the independent or predictor variables?
Several typos:
Table1: "Edad" to "Age".
Table 4: "Ansiety" should be "Anxiety".
Conclusion, line 341, "CO-VID-19" should be "COVID-19".

Author Response

(The authors gave the same response as above.)
